# Assessment of the Safety Profile of Chimeric Marker Vaccine against Classical Swine Fever: Reversion to Virulence Study

**DOI:** 10.3390/v16071120

**Published:** 2024-07-12

**Authors:** Loc Tan Huynh, Mikihiro Otsuka, Maya Kobayashi, Hung Dinh Ngo, Lim Yik Hew, Takahiro Hiono, Norikazu Isoda, Yoshihiro Sakoda

**Affiliations:** 1Laboratory of Microbiology, Department of Disease Control, Faculty of Veterinary Medicine, Hokkaido University, Sapporo 060-0818, Japan; huynhtanloc@vetmed.hokudai.ac.jp (L.T.H.); jya1225kuchu@eis.hokudai.ac.jp (M.K.); microbio-data@vetmed.hokudai.ac.jp (H.D.N.); ylhew@vetmed.hokudai.ac.jp (L.Y.H.); hiono@vetmed.hokudai.ac.jp (T.H.); nisoda@vetmed.hokudai.ac.jp (N.I.); 2Faculty of Veterinary Medicine, College of Agriculture, Can Tho University, Can Tho 900000, Vietnam; 3The Gifu Hida Livestock Hygiene Service Center, Gifu 506-8688, Japan; otsuka-mikihiro@pref.gifu.lg.jp; 4One Health Research Center, Hokkaido University, Sapporo 060-0818, Japan; 5International Collaboration Unit, International Institute for Zoonosis Control, Hokkaido University, Sapporo 001-0020, Japan; 6Hokkaido University Institute for Vaccine Research and Development (HU-IVReD), Hokkaido University, Sapporo 001-0021, Japan

**Keywords:** chimeric marker vaccine, safety profile, classical swine fever virus, pestivirus

## Abstract

Chimeric marker vaccine candidates, vGPE^−^/PAPeV E^rns^ and vGPE^−^/PhoPeV E^rns^, have been generated and their efficacy and capability to differentiate infected from vaccinated animals were confirmed in previous studies. The safety profile of the two chimeric marker vaccine candidates, particularly in the potential reversion to virulence, was evaluated. Each virus was administered to pigs with a dose equivalent to the vaccination dose, and pooled tonsil homogenates were subsequently inoculated into further pigs. Chimeric virus vGPE^−^/PAPeV E^rns^ displayed the most substantial attenuation, achieving this within only two passages, whereas vGPE^−^/PhoPeV E^rns^ was detectable until the third passage and disappeared entirely by the fourth passage. The vGPE^−^ strain, assessed alongside, consistently exhibited stable virus recovery across each passage without any signs of increased virulence in pigs. In vitro assays revealed that the type I interferon-inducing capacity of vGPE^−^/PAPeV E^rns^ was significantly higher than that of vGPE^−^/PhoPeV E^rns^ and vGPE^−^. In conclusion, the safety profile of the two chimeric marker vaccine candidates was affirmed. Further research is essential to ensure the stability of their attenuation and safety in diverse pig populations.

## 1. Introduction

Classical swine fever (CSF), a highly contagious and often fatal disease, poses a significant threat to domestic pigs and wild boars worldwide [1,2]. Its causative agent, CSF virus, belongs to the *Pestivirus* genus within the *Flaviviridae* family. The CSFV genome, approximately 12.3 kb in length, comprises a single large open reading frame (ORF) flanked by untranslated regions (UTRs) at the 5′ and 3′ ends. This ORF encodes a polyprotein of approximately 4000 amino acids which undergoes cleavage by cellular and viral proteases to yield twelve proteins, including structural proteins (C, E^rns^, E1, and E2) and nonstructural proteins (N^pro^, p7, NS2, NS3, NS4A, NS4B, NS5A, and NS5B) [3]. Regarding the role of nonstructural proteins, it has been determined that the minimal set essential for pestiviral RNA replication consists of NS3 to NS5 [3]. Additionally, the unique proteins of pestiviruses are N^pro^ and E^rns^, which are recognized for their roles in suppressing the innate immune response [4]. In particular, the glycoprotein E^rns^ is unique to pestiviruses, lacking homologs even among closely related hepaciviruses or pegiviruses. This protein exhibits distinctive characteristics, including an unusual membrane anchor, secretion as a protein, and ribonuclease (RNase) activity [3].

Vaccination is a crucial strategy for controlling CSF outbreaks, and live attenuated vaccines (LAVs) are commonly employed due to their efficacy [1,5]. However, LAVs carry inherent risks, including the potential for reversion to virulence, where the vaccine strain regains its ability to cause the re-emergence of CSF after introducing the vaccine in the field [6,7]. A previous study cautioned against the reversion to virulence of the LAV, FlagT4v, whose virulence was restored during five successive passages in piglets [8]. This risk is further heightened by a recent study, which demonstrated that a chimeric CSFV vSM/CE2 containing the E2 gene of the vaccine C-strain on the genetic background of the virulent CSFV strain Shimen was reversed to virulence after serial passages in PK15 cells, and the increased virulence of its mutants was also confirmed in the experimental infection in pigs [9]. Despite the extensive utilization of the LAV GPE^−^ strain in Asian and Pacific regions for numerous years [10,11], the confirmed risk of reversion to virulence of LAV GPE^−^ after 11 serial passages in pigs has been affirmed in a previous study [12], underscoring the importance of assessing the safety profile, particularly in vaccine reversion to virulence [11].

The structural glycoprotein E^rns^, which is unique to pestiviruses and an essential component of the viral envelope, has multifaceted functions critical for viral replication, pathogenesis, and immune evasion [1,3]. E^rns^ possesses RNase activity, enabling it to degrade viral and host RNAs and thereby suppressing the host’s innate immune response and facilitating viral persistence within the host. Despite its crucial functions in viral biology, E^rns^ also plays a significant role as an antigen in diagnosing and developing genetically engineered vaccines. Considering that the ability of E^rns^ to elicit neutralizing antibodies is not the primary component [13,14] and is a suitable candidate for epitope substitution [15], several chimeric pestiviruses possessing the E^rns^ which is genetically related within the *Pestivirus* genus, have been intensively developed to improve CSF marker vaccine candidates with the potential to combine the efficacy of LAVs and differentiating infected from vaccinated animal (DIVA) properties [16,17,18]. Nevertheless, additional assessments of these candidates, covering aspects such as safety profiles, particularly in the potential for reversion to virulence, have not yet been conducted [16,17,18]. Notably, a recent study has confirmed the reversion to virulence associated with the CSFV E^rns^ glycoprotein [19], in which the recombinant virus C-W300G derived from the virulent CSFV Alfort/Tübingen strain, possessing a mutant that eliminated the RNase activity of E^rns^, initially demonstrated attenuation in pigs; however, subsequent inoculation of the reisolated virus in pigs revealed a rapid reversion to virulence and restored the wild-type characteristics. In this scenario, the insufficiency of safety assessments raises concerns about the possibility of regaining virulence, especially concerning chimeric pestivirus vaccines, designed to express foreign genes from related pestiviruses [11].

Previous studies have examined two chimeric marker vaccine candidates, vGPE^−^/PAPeV E^rns^ and vGPE^−^/PhoPeV E^rns^, generated based on the backbone of the LAV vGPE^−^ strain [12,20]. The chimeric marker vaccine candidates possess the glycoprotein E^rns^ of Pronghorn antelope pestivirus (PAPeV) [21] and Phocoena pestivirus (PhoPeV) [22], respectively, which are distantly related to CSFV. Their vaccine efficacy and DIVA capability were also confirmed in previous studies [18,23]. Although serial passages in SK-L cells confirmed high viral growth and genetic stability [18], their safety characteristics have not been validated through serial passages in pigs. This study continued a series of research projects aimed at evaluating the safety profile of the two chimeric marker vaccine candidates, particularly in the potential reversion to virulence.

## 2. Materials and Methods

### 2.1. Cells and Viruses

The swine kidney line-L (SK-L) cells [24] were cultured in Eagle’s minimum essential medium (EMEM; Nissui Pharmaceutical, Tokyo, Japan) supplemented with 0.295% tryptose phosphate broth (Becton Dickinson, Franklin Lakes, NJ, USA), 10 mM N,N-bis-(2-hydroxyethyl)-2-aminoethanesulfonic acid (BES; Sigma-Aldrich, St. Louis, MO, USA), sodium bicarbonate (Nacalai Tesque, Kyoto, Japan), and 10% horse serum (HS; Thermo Fisher Scientific, Waltham, MA, USA). SK-L cells were employed to evaluate viral growth kinetics, virus titration, and serological testing. Cultured cells were incubated at 37 °C with 5% CO_2_. The SK6-MxLuc cell line used in the IFN bioassay, harboring an Mx/Luc reporter gene, was cultured in EMEM supplemented with 10 mM BES and 7% HS [25,26].

An engineered clone of CSFV-modified LAV (vGPE^−^) derived from pGPE^−^ [12,20] was used. Two chimeric marker vaccine candidates vGPE^−^/PAPeV E^rns^ and vGPE^−^/PhoPeV E^rns^, possessing PAPeV E^rns^ and PhoPeV E^rns^, respectively, were generated and evaluated in previous studies [18,23].

### 2.2. Virus Titration

To determine viral titers, blood samples or 10% organ homogenates were cultured with confluent SK-L cells in 6-well plates. Serially diluted 10-fold samples (either blood samples or 10% organ homogenates) were cultured simultaneously in 96-well plates. After 96 h incubation at 37 °C in 5% CO_2_, cells were air-dried, heat-fixed, and subsequently stained with anti-NS3 monoclonal antibody 46/1 as the primary antibody for immunoperoxidase staining [27]. Viral titers were calculated using the Reed and Muench method and displayed as 50% tissue culture infective dose per milliliter (TCID_50_/mL) [28].

### 2.3. Viral Growth Kinetics

The growth kinetics of the vGPE^−^ virus and the two chimeric marker vaccine candidates in SK-L cells were assessed by introducing them to confluent cell monolayers at a multiplicity of infection (MOI) of 0.001. After inoculation, SK-L cells were cultured at 37 °C with 5% CO_2_. Cell supernatants were collected at 0, 1, 2, 3, 4, 5, 6, and 7 days postinoculation (dpi). Viral titers were determined in triplicate using SK-L cells, according to the abovementioned virus titration.

### 2.4. Bioassay for Type I IFN Measurement

The bioassay was conducted following previously established methods for measuring swine IFN-α/β [25,26]. In summary, supernatants from SK-L cells infected with each virus were rendered inactive using an ultraviolet cross-linker (DNA-FIX DF-254; ATTO, Tokyo, Japan) before being introduced to SK6-MxLuc cells. The recombinant swine IFN-*α* produced in a previous study was employed as the standard [29]. Cell lysates were generated using 100 µL passive lysis buffer, and firefly luciferase activities were quantified using the Dual-Luciferase Reporter Assay System and POWERSCAN^®^4 (Agilent Technologies International Japan Ltd., Tokyo, Japan). Results were documented across three independent trials, each executed in duplicate.

### 2.5. Sequencing

The entire genome of the recovered vGPE^−^ virus or each chimeric marker vaccine candidate passaged in pigs was verified as described previously [12,18]. In short, the BigDye Terminator version 3.1 Cycle Sequencing Kit (Thermo Fisher Scientific) and an ABI 3500 Genetic Analyzer (Thermo Fisher Scientific) were utilized for nucleotide sequencing of polymerase chain reaction (PCR) fragments of viral cDNA derived from viral RNA. Sequencing data were processed using GENETYX^®^ Network edition version 15.0.1 (GENETYX, Tokyo, Japan).

### 2.6. Animal Use

Pigs were purchased from a CSFV-free farm (Yamanaka Chikusan, Hokkaido, Japan) in Hokkaido and confirmed to be free of antibodies against CSFV, as outlined in a previous study [18].

### 2.7. Reversion to Virulence Study through Experimental Infection in Pigs

The safety test involved serial passage of viruses in pigs, as illustrated in Figure 1. Briefly, either each chimeric marker vaccine candidate (vGPE^−^/PAPeV E^rns^ or vGPE^−^/PhoPeV E^rns^) or vGPE^−^ was introduced via intramuscular injection of 2 mL cell culture supernatant of each virus into two 2-week-old crossbred Landrace × Duroc × Yorkshire SPF pigs (Yamanaka Chikusan) per passage at a dose of 10^4.0^ TCID_50_, recommended as equivalent to one dose of the vaccine in a previous study [18]. Twenty-two pigs were employed in this study. Among them, 4 pigs were injected with vGPE^−^/PAPeV E^rns^ within two passages, 8 pigs were injected with vGPE^−^/PhoPeV E^rns^ within four passages, and 10 pigs were injected with vGPE^−^ within five passages. All pigs were monitored daily for clinical signs according to a scoring system for 7 days [30]. Blood samples were collected in tubes containing EDTA (Venoject II VP-NA050K; Terumo, Tokyo, Japan) at 0, 2, 3, 4, 5, 6, and 7 dpi. Serum samples were collected using tubes containing a blood coagulation factor (Venoject II VP-P075K; Terumo) at 0 and 7 dpi. The levels of CSFV-specific neutralizing antibodies of pigs at 0 and 7 dpi were evaluated by a serological test. All survival pigs were euthanized at 7 dpi. Tonsil samples were collected aseptically. For virus titration, 10% organ homogenates were prepared and stored at –80 °C. Virus titers were measured and displayed as TCID_50_/mL (blood) or TCID_50_/g (tissue). If no virus recovery was found after each passage, the experiment was ended. If virus recovery was confirmed, pooled tonsil homogenates were used to inoculate the next two pigs of the same age and origin by the same route as before. The serial passage in pigs was carried out five times in total from the start of the first vaccination experiment.

### 2.8. Serum Neutralization Test (SNT)

The assay was conducted by employing the luciferase-based method, as outlined previously [31]. In summary, an equal volume of serum and 100 TCID_50_ of CSFV vGPE^−^/HiBiT [32] were thoroughly mixed and incubated at 37 °C for 1 h. The mixture and SK-L cell suspension were cultured in 96-well plates and incubated at 37 °C with 5% CO_2_ for 96 h. Neutralizing antibody titers were determined through the luciferase assay utilizing the Nano-Glo HiBiT Lytic Detection System (Promega, Madison, WI, USA) and POWERSCAN^®^4.

### 2.9. Statistical Analysis

Statistical analyses of the data were conducted using Microsoft Excel 365 (Microsoft Corp., Redmond, WA, USA). One-way analysis of variance was performed, followed by Student’s *t*-test with Bonferroni correction.

### 2.10. Ethics Statement

Animal experiments were approved by the Institutional Animal Care and Use Committee of the Faculty of Veterinary Medicine, Hokkaido University (approval no. 18-0038, approved on 26 March 2018 and approval no. 23-0029, approved on 23 March 2023), and performed according to the guidelines of this committee. Animals reaching the humane endpoint were euthanized through intracardial injection of thiopental sodium (Ravonal^®^; Nipro ES Pharma Co., Ltd., Osaka, Japan) after deep sedation with isoflurane (Fujifilm Wako Pure Chemical Co., Ltd., Osaka, Japan), as described previously [18]. The experiments took place in animal facilities accredited by the Association for Assessment and Accreditation of Laboratory Animal Care International (AAALAC International).

## 3. Results

### 3.1. Attenuation of Chimeric Viruses during Serial Passages in Pigs

To determine whether the two chimeric marker vaccine candidates, vGPE^−^/PAPeV E^rns^ or vGPE^−^/PhoPeV E^rns^, and the vGPE^−^ strain can regain pathogenicity in pigs through forced transmission from pig to pig, the virus underwent serial passages in pigs via the injection of the virus recovered from inoculated animals. As a result, the passaging of each virus seed and tonsil homogenates from vaccinated pigs was performed 2, 4, and 5 times for vGPE^−^/PAPeV E^rns^, vGPE^−^/PhoPeV E^rns^, and vGPE^−^, respectively (Figure 1). Both chimeric marker vaccine candidates vGPE^−^/PAPeV E^rns^ and vGPE^−^/PhoPeV E^rns^ complied with the test due to the absence of virus recovery after the second and fourth passages, respectively. However, the vGPE^−^ virus was still recovered until the fifth passage in pigs.

In each passage in pigs, the clinical signs of inoculated pigs were monitored daily until 7 dpi, and virus recovery in blood samples was determined daily between days 2 and 7 during the experiment, as shown in Table 1. Based on a clinical scoring system [30], there were no abnormalities in clinical signs in all pigs of each passage. The results of the SNT also confirmed the absence of neutralizing antibodies against CSFV in the naïve pigs at 0 and 7 dpi. The absence of virus recovery in blood (Table 1) samples in each passage in pigs inoculated with either vGPE^−^/PAPeV E^rns^ or vGPE^−^/PhoPeV E^rns^ was evidenced by virus titration. Transient viremia was confirmed in the second passage in one pig inoculated with the recovered P1/vGPE^−^ virus at 6 dpi. Although viremia was undetectable in pigs inoculated with the recovered vGPE^−^ virus from the second to the fourth passage, the presence of the virus was reconfirmed in the fifth passage. In detail, virus recovery was confirmed positive in 6-well plates but lower than the detection limit of 10^0.8^ TCID_50_/mL in 96-well plates in blood samples at 5, 6, and 7 dpi in two pigs inoculated with the recovered P4/vGPE^−^ virus. At 7 dpi in each passage, pigs were euthanized, and virus recovery was then determined in tonsil homogenates. After the first passage, virus titers in the tonsils of inoculated pigs were <10^3.0^ TCID_50_/g (Table 1) in one pig inoculated with either vGPE^−^/PAPeV E^rns^ or vGPE^−^/PhoPeV E^rns^; meanwhile, virus recovery was confirmed positive in 6-well plates but lower than the detection limit of 10^1.8^ TCID_50_/g in 96-well plates in the remaining pig in each group. Virus recovery was undetectable in tonsil homogenates after the second passage in two pigs inoculated with P1/vGPE^−^/PAPeV E^rns^ and one pig inoculated with P1/vGPE^−^/PhoPeV E^rns^. Although the remaining pig inoculated with P1/vGPE^−^/PhoPeV E^rns^ was confirmed with the virus recovery in 6-well plates, the virus titer was lower than the detection limit of 10^1.8^ TCID_50_/g in 96-well plates. This occurrence was also confirmed in the third passage of pigs inoculated with the recovered P2/vGPE^−^/PhoPeV E^rns^ virus. However, no virus recovery was confirmed at the fourth passage in pigs inoculated with the P3/vGPE^−^/PhoPeV E^rns^ virus. In contrast, virus recovery in the tonsils of pigs inoculated with recovered vGPE^−^ virus persisted (10^3.1^–10^5.0^ TCID_50_/g) across the first to the fifth passage. These results indicated that two chimeric marker vaccine candidates, vGPE^−^/PAPeV E^rns^ and vGPE^−^/PhoPeV E^rns^, were attenuated through serial passages in pigs; meanwhile, the vGPE^−^ showed replication stability in pigs without clinical signs.

### 3.2. Amino Acid Substitutions of Chimeric Viruses Passaged in Pigs

To ascertain whether serial passages of the two chimeric marker vaccine candidates and vGPE^−^ in pigs led to the selection of mutant viruses, the entire genome sequences of the recovered viruses after each passage were examined via direct sequencing of DNA fragments amplified via reverse transcription–PCR, originating from viral RNA. The amino acid substitutions were identified in the viruses recovered after each passage, as shown in Figure 2. Two amino acid substitutions in C protein (L264P) and E1 protein (M660T) were found after the first passage in pigs inoculated with vGPE^−^/PAPeV E^rns^. The substitutions were confirmed in the NS5B protein of the P3/vGPE^−^/PhoPeV E^rns^, where the substitutions were identified as asparagine to aspartic acid and threonine to isoleucine at positions 3289 and 3842, respectively. In NS5B, threonine to isoleucine substitution at position 3842 arose after the second passage and persisted until the third. After five passages, the P5/vGPE^−^ virus exhibited a similar threonine to isoleucine substitution at position 3842, as observed in the P2- and P3/vGPE^−^/PhoPeV E^rns^ (Figure 2). The P5/vGPE^−^ virus demonstrated quasispecies diversity, indicated by the appearances of additional amino acids resided at L1277M, L2621S, and K3418E, corresponding to the NS2, NS4B, and NS5B proteins, respectively. Amino acid sequences of the recovered viruses, P1/vGPE^−^/PAPeV E^rns^, P3/vGPE^−^/PhoPeV E^rns^, or P5/vGPE^−^, were compared to those of vGPE^−^ and GPE^−^ vaccine strains, and its parental CSFV ALD strain and recombinant vALD-A76, which was generated from an infectious cDNA clone of the virulent strain ALD-A76 in a previous study [29] and has a different sequence compared to the parental ALD strain (Figure 2). According to the sequence alignment of amino acids, T660 (E1) in P1/vGPE^−^/PAPeV E^rns^ and I3842 (NS5B) in P3/vGPE^−^/PhoPeV E^rns^ and P5/vGPE^−^ were identical to the parental CSFV ALD strain and vALD-A76.

### 3.3. In Vitro Growth Kinetics of vGPE^−^ and Chimeric Viruses

To assess the impact of substituting the glycoprotein E^rns^ on viral growth kinetics and IFN-α/β production in swine cells, multistep growth curves of the two chimeric marker vaccine candidates and vGPE^−^ in SK-L cells at 37 °C were investigated. In Figure 3, vGPE^−^/PAPeV E^rns^ exhibited a higher replication tendency compared to vGPE^−^/PhoPeV E^rns^ but was nearly equivalent to vGPE^−^, except for higher replication levels than vGPE^−^ at 1 and 6 dpi. Meanwhile, vGPE^−^/PhoPeV E^rns^ propagated to nearly the same extent as vGPE^−^ but at a slower rate than vGPE^−^ at 1 and 5 dpi. Despite the growth kinetics of the two chimeric viruses being nearly comparable to that of vGPE^−^, significant differences were observed in IFN-α/β induction when SK-L cells were infected with each virus. Previous studies demonstrated the robust IFN induction capacity of vGPE^−^ [20,26], which also corroborated the IFN induction of vGPE^−^ in this study. Of particular interest is the observation that the IFN-producing capacity of vGPE^−^/PAPeV E^rns^ surpassed that of vGPE^−^. A large amount of IFN production by vGPE^−^/PAPeV E^rns^ persisted from 3 dpi onward. Similarly, vGPE^−^/PhoPeV E^rns^ exhibited higher IFN induction than vGPE^−^; however, this discrepancy was not evident at 3 and 4 dpi. Between the two chimeric virus strains, although vGPE^−^/PhoPeV E^rns^ showed a higher IFN level compared with vGPE^−^/PAPeV E^rns^ at 1 dpi, significant differences in IFN induction were noted from 2 to 6 dpi, with vGPE^−^/PAPeV E^rns^ consistently eliciting higher IFN levels than vGPE^−^/PhoPeV E^rns^.

## 4. Discussion

In Japan, the LAV GPE^−^ strain was developed and extensively utilized to control CSF. The GPE^−^ strain originated from the highly virulent ALD strain, undergoing multiple passages and biological cloning in swine testicle cells, bovine testicle cells, and primary guinea pig kidney cells [33]. Field studies demonstrated the safety and efficacy of the GPE^−^ vaccine [34]. A reverse genetic system for the GPE^−^ strain was recently established using its infectious cDNA clone. Consequently, the vGPE^−^ strain, harboring 10 amino acid substitutions in comparison to the GPE^−^ strain [35], was created to aid in understanding CSF pathogenesis [12,20,26,36], supporting the quality control of the GPE^−^ vaccine. Recently, two chimeric candidates, vGPE^−^/PAPeV E^rns^ and vGPE^−^/PhoPeV E^rns^, were synthesized in the joint effort toward marker vaccine development [18]. These candidates were derived from the LAV vGPE^−^ vaccine strain as a backbone and possessed the glycoprotein E^rns^ sourced from distantly related pestiviruses. Considering the safety assessment according to the previous reports on the virulence gain observed with the GPE^−^ vaccine strain [12], there arose concerns regarding the utilization of the vGPE^−^ strain as a backbone and the incorporation of foreign epitopes into LAV, which led to questions about their stability and safety profiles. To better understand these concerns, this study showed that the attenuation of the two chimeric marker vaccine candidates can be affirmed by substituting the CSFV glycoprotein E^rns^ with either PAPeV E^rns^ or PhoPeV E^rns^. Meanwhile, the vGPE^−^ strain maintained stable replication without increasing virulence in pigs.

Throughout the serial passages in pigs, the chimeric marker vaccine candidate vGPE^−^/PAPeV E^rns^ exhibited the highest attenuation level, achieving this within just two passages. Although the recovery of vGPE^−^/PhoPeV E^rns^ was detectable until the third passage, in which the virus recovery was either below the detection threshold or not present, it completely disappeared by the fourth, affirming its attenuation. The vGPE^−^ strain, which serves as the backbone for the two chimeric viruses, demonstrated stable virus recovery across all passages without indicating increased virulence in pigs, suggesting potential advantages in terms of safety and efficacy, as confirmed in a previous study [20]. This is attributed to its development through a reverse genetic system-based production method [12,16,26,35,36,37], ensuring strict quality management for the vaccine. Meanwhile, the changes acquired during passages in pigs inoculated with GPE^−^ were attributed to the vaccine inoculum harboring a quasispecies population and the experimental design of serial passage in pigs involved in administering a high inoculation dose (10^7.6^ TCID_50_) of the GPE^−^ virus at the first passage, and tonsil homogenates were then collected at the peak of virus recovery at 4 dpi, as confirmed previously [12], which differed from the design in this study.

The recovered viruses in each passage were utilized to analyze amino acid substitutions in their genomes. Surprisingly, amino acid substitutions in the structural proteins (C and E1) were identified in P1/vGPE^−^/PAPeV E^rns^, even though the virus disappeared in the second passage. Given that the last eight residues (262–267) at the C-terminus of the C protein significantly influence the proteasomal degradation of CSFV C protein and determine the cleavage efficiency of the C protein by signal peptide peptidase, as established previously [38], the P264L substitution at the C-terminus of the C protein, observed in the P1/vGPE^−^/PAPeV E^rns^, could potentially impact efficient viral replication through interactions with both viral and cellular proteins [39]. The M660T substitution observed in the E1 protein of P1/vGPE^−^/PAPeV E^rns^ was found to be identical to the ancestral highly virulent ALD strain of GPE^−^ and vGPE^−^. Similarly, amino acid homology with the ALD strain was also identified in P2 and P3/vGPE^−^/PhoPeV E^rns^ at residue I3842. Among the two amino acid substitutions in the nonstructural protein (NS5B) of P3/vGPE^−^/PhoPeV E^rns^, the N3289D substitution was situated in the N-terminal domain (NTD), while another substitution (T3842I) resided in the thumb domain of NS5B. Because the thumb domain within the CSFV NS5B protein had a discriminating role, precisely regulating access to the active site during initiation by restricting entry to the template-binding site [40], the T3842I substitution, which emerged in P2/vGPE^−^/PhoPeV E^rns^ and persisted in P3/vGPE^−^/PhoPeV E^rns^, was identical to ALD and vALD-A76 strains and might enhance viral replication, which was also found at the same position in NS5B of P5/vGPE^−^. However, the subsequent discovery of N3289D in the NTD in NS5B protein of P3/vGPE^−^/PhoPeV E^rns^ raised the hypothesis that the impaired replication observed in P4/vGPE^−^/PhoPeV E^rns^ could result from disrupting the catalytic activity of the RNA-dependent RNA polymerase [40,41]. Meanwhile, the quasispecies population was identified by the occurrence of amino acid changes in nonstructural proteins (NS2, NS4B, and NS5B) in P5/vGPE^−^, essential for pestiviral RNA replication [3]. Given the attenuation observed in pigs for both chimeric viruses, the present study did not comprehensively investigate the functional implications of these mutations. However, further studies can elucidate viral attenuation via these amino acid substitutions to understand viral pathogenesis.

Given the attenuation of the two chimeric marker vaccine candidates in pigs, their capacities for IFN-α/β induction were investigated in vitro. Pigs inoculated with the highly IFN-α/β-inducing vGPE^−^/PAPeV E^rns^ strain exhibited viral attenuation within two passages. A similar phenomenon was observed in pigs injected with the IFN-α/β-inducing vGPE^−^/PhoPeV E^rns^ strain, where the virus became undetectable within four passages. Although the IFN-α/β induction by vGPE^−^ demonstrated a comparable pattern to that of vGPE^−^/PhoPeV E^rns^ in vitro, virus recovery persisted in the tonsils in all pigs until the fifth passage. Previous investigations into the role of E^rns^ as an IFN antagonist primarily focused on evaluating the RNase activity of E^rns^ from bovine viral diarrhea virus (BVDV) and CSFV, which provided indirect evidence suggesting comparable RNase activities between E^rns^ from BVDV and CSFV [42,43]. A recent study expanded upon this research by assessing the RNase activity of E^rns^ from nearly all viruses classified within the *Pestivirus* genus, including PAPeV E^rns^ and PhoPeV E^rns^ [4]. Notably, the RNase activity of E^rns^ from both PAPeV and PhoPeV was found to be the least active, exhibiting ~20- to 30-fold lower activity than BVDV [4]. It might indirectly support the highly IFN-α/β-inducing vGPE^−^/PAPeV E^rns^ strain that showed attenuation in pigs in this study. Despite the lack of correlation between the in vivo characteristics of vGPE^−^/PhoPeV E^rns^ and its IFN-α/β induction profile, the mechanism underlying its attenuation in pigs remained elusive. Further research should be performed to elucidate this mechanism comprehensively.

While the current study showed that the two chimeric viruses were attenuated within fewer than five passages, comprehensive studies are still required to fully establish the safety profiles of these chimeric viruses in the future. This includes following the World Organisation for Animal Health recommendations [44], such as considering the size and age of the pigs used in the experiments and extending a few more passages even after the virus becomes undetectable to assess the stability of the attenuation and the potential for reversion to virulence, ensuring it meets the life expectancy of breeder pigs. In addition, it is essential to consider the potential for latent infection by the chimeric virus, where the virus can remain dormant within the host without causing symptoms, potentially reactivating later or persisting in offspring. Therefore, future research should prioritize investigating the safety of chimeric viruses in breeding stocks (sows or boars) through long-term monitoring, especially in pregnant sows, to mitigate the risk of transplacental transmission in practical contexts. These assessments will ensure a more accurate and practical safety evaluation, aligning with field conditions that better reflect real-world settings.

In summary, this study examined the safety profile of the two chimeric marker vaccine candidates, vGPE^−^/PAPeV E^rns^ and vGPE^−^/PhoPeV E^rns^, which were derived from the LAV vGPE^−^ vaccine strain and carried the glycoprotein E^rns^ from distantly related pestiviruses. Through serial passages in pigs, both chimeric viruses exhibited an attenuation, suggesting their potential as safety vaccine candidates. These findings underscored the promise of these chimeric marker vaccine candidates for CSF control and eradication strategies. Since vGPE^−^/PAPeV E^rns^ exhibited a high attenuation level in pigs and the persistence of vGPE^−^/PhoPeV E^rns^ was confirmed up to the third passage, additional investigations involving a larger cohort of animals must be conducted to elucidate the underlying mechanisms driving its attenuation. Further rigorous evaluations are warranted to comprehensively assess their stability and safety profiles in diverse pig populations and ensure their long-term effectiveness and safety in the field.

## Figures and Tables

**Figure 1 viruses-16-01120-f001:**
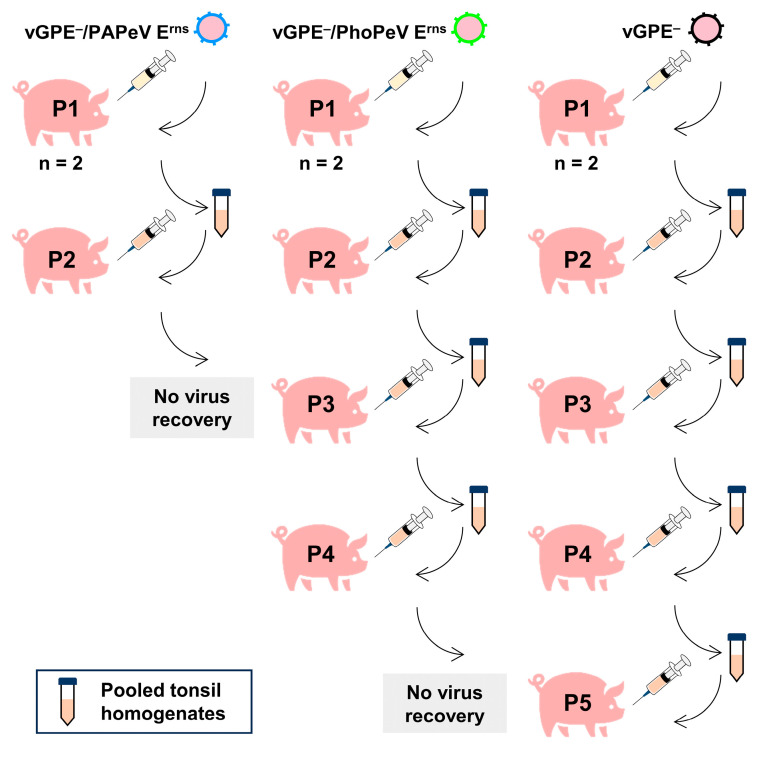
Study design for serial passages of viruses in pigs. Virus seeds, including vGPE^−^/PAPeV E^rns^, vGPE^−^/PhoPeV E^rns^, and vGPE^−^, were independently inoculated into two pigs per passage (P). A total of four pigs received injections of vGPE^−^/PAPeV E^rns^ over 2 passages (**left** side). Another eight pigs were injected with vGPE^−^/PhoPeV E^rns^ over 4 passages (**center**). Ten pigs were injected with vGPE^−^ over 5 passages (**right** side).

**Figure 2 viruses-16-01120-f002:**
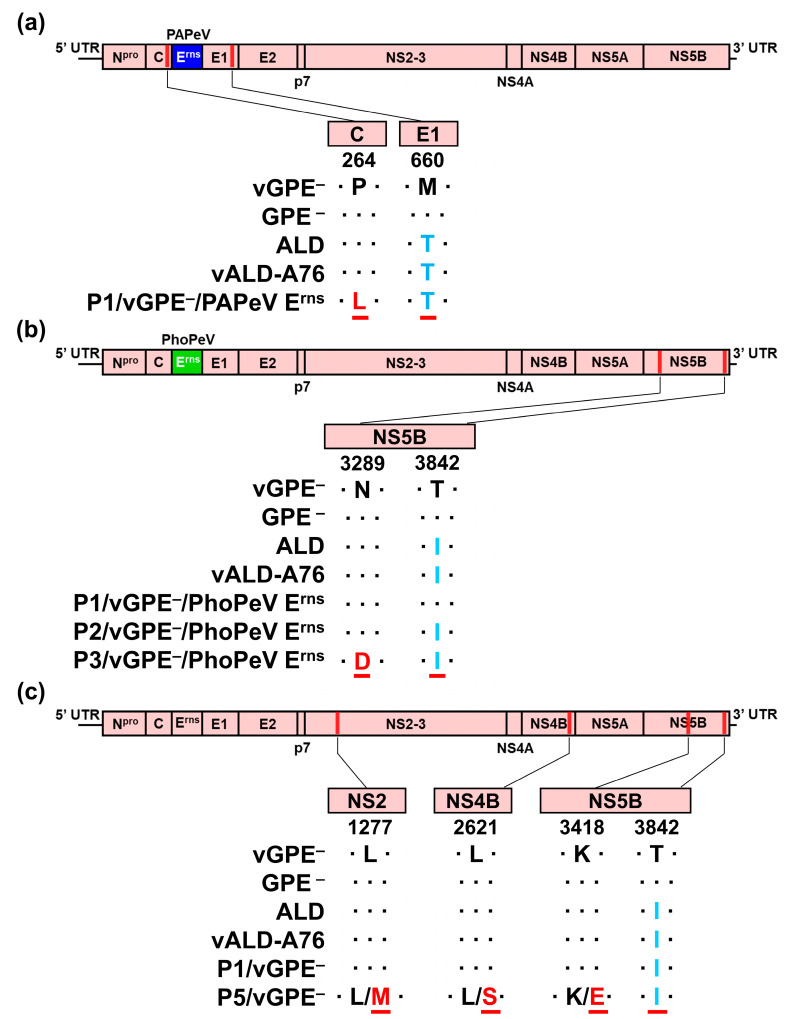
Schematic representation of amino acid substitutions among CSFV strains and passaged viruses in pigs. (**a**) Two amino acid substitutions (P264L and M660T) were found in the P1/vGPE^−^/PAPeV E^rns^. (**b**) Two amino acid substitutions (N3289D and T3842I) were found in the P3/vGPE^−^/PhoPeV E^rns^. (**c**) The substitution of threonine to isoleucine at position 3842 was identified in P5/vGPE^−^. The P5/vGPE^−^ virus also exhibited quasispecies diversity with the emergence of amino acid changes at residues L1277M, L2621S, and K3418E. The methionine encoded by the AUG start codon is defined as position 1, and the subsequent residues are identified according to the backbone vGPE^−^ strain. Two amino acids were found (quasispecies). The periods indicate the identity with the sequence of virus seeds, vGPE^−^/PAPeV E^rns^, vGPE^−^/PhoPeV E^rns^, or vGPE^−^. The amino acid sequences different from vGPE^−^ are underlined in red. The amino acid sequences unique to vGPE^−^ and GPE^−^ strains are indicated as periods. The amino acid sequence unique to the parental CSFV ALD and vALD-A76 strains is highlighted in light blue characters.

**Figure 3 viruses-16-01120-f003:**
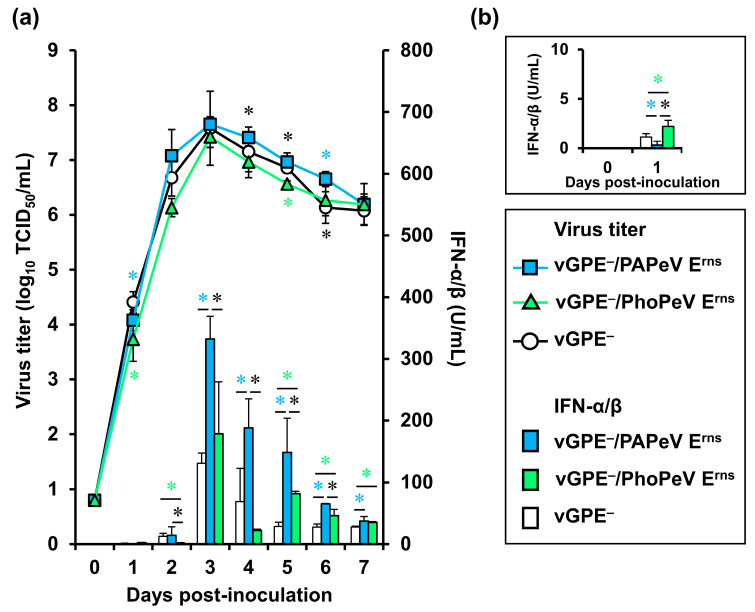
In vitro growth kinetics and IFN-α/β induction in SK-L cells inoculated with vGPE^−^/PAPeV E^rns^, vGPE^−^/PhoPeV E^rns^, or vGPE^−^. (**a**) SK-L cells were inoculated with vGPE^−^/PAPeV E^rns^, vGPE^−^/PhoPeV E^rns^, or vGPE^−^ at an MOI of 0.001 and incubated at 37 °C with 5% CO_2_. Cell supernatants were collected at 0, 1, 2, 3, 4, 5, 6, and 7 dpi. Virus titers were measured and displayed as 50% TCID_50_/mL. The IFN-α/β bioactivity in the SK-L cell supernatants was measured using the SK6-MxLuc cells. (**b**) The bar chart scale depicting IFN-α/β induction at 0 and 1 dpi was enlarged for better visualization and interpretation of data. Error bars represent the standard deviations. The significance of viral growth differences was indicated. The blue, green, and black asterisks indicate *p* < 0.05 between vGPE^−^ and vGPE^−^/PAPeV E^rns^, vGPE^−^ and vGPE^−^/PhoPeV E^rns^, vGPE^−^/PAPeV E^rns^ and vGPE^−^/PhoPeV E^rns^, respectively.

**Table 1 viruses-16-01120-t001:** Virus recovery from blood and tissue samples in pigs inoculated with vGPE^−^/PAPeV E^rns^, vGPE^−^/PhoPeV E^rns^, or vGPE^−^.

Passage (P)	Virus	Pig ID	Clinical Signs	Virus Recovery								SNT Titer at dpi	
				Blood (log_10_ TCID_50_/mL) at dpi							Tissue (log_10_ TCID_50_/g)		
				0	2	3	4	5	6	7	Tonsil	0	7
P1	vGPE^−^/PAPeV E^rns^	#384	N/O	—	—	—	—	—	—	—	10^2.8^	<1	<1
		#385	N/O	—	—	—	—	—	—	—	+	<1	<1
	vGPE^−^/PhoPeV E^rns^	#386	N/O	—	—	—	—	—	—	—	10^2.3^	<1	<1
		#387	N/O	—	—	—	—	—	—	—	+	<1	<1
	vGPE^−^	#382	N/O	—	—	—	—	—	—	—	10^3.1^	<1	<1
		#383	N/O	—	—	—	—	—	—	—	10^4.6^	<1	<1
P2	vGPE^−^/PAPeV E^rns^	#390	N/O	—	—	—	—	—	—	—	—	<1	<1
		#391	N/O	—	—	—	—	—	—	—	—	<1	<1
	vGPE^−^/PhoPeV E^rns^	#392	N/O	—	—	—	—	—	—	—	—	<1	<1
		#393	N/O	—	—	—	—	—	—	—	+	<1	<1
	vGPE^−^	#388	N/O	—	—	—	—	—	+	—	10^3.9^	<1	<1
		#389	N/O	—	—	—	—	—	—	—	10^4.0^	<1	<1
P3	vGPE^−^/PhoPeV E^rns^	#396	N/O	—	—	—	—	—	—	—	—	<1	<1
		#397	N/O	—	—	—	—	—	—	—	+	<1	<1
	vGPE^−^	#394	N/O	—	—	—	—	—	—	—	10^4.0^	<1	<1
		#395	N/O	—	—	—	—	—	—	—	10^4.0^	<1	<1
P4	vGPE^−^/PhoPeV E^rns^	#405	N/O	—	—	—	—	—	—	—	—	<1	<1
		#406	N/O	—	—	—	—	—	—	—	—	<1	<1
	vGPE^−^	#403	N/O	—	—	—	—	—	—	—	10^4.8^	<1	<1
		#404	N/O	—	—	—	—	—	—	—	10^3.6^	<1	<1
P5	vGPE^−^	#407	N/O	—	—	—	—	—	—	+	10^5.0^	<1	<1
		#408	N/O	—	—	—	—	+	+	—	10^4.8^	<1	<1

N/O, not observed; —, not isolated in a 6-well plate; +: isolated in a 6-well plate and was lower than the detection limit of TCID_50_ (10^0.8^ TCID_50_/mL for blood samples and 10^1.8^ TCID_50_/g for tissue samples) in a 96-well plate.

## Data Availability

Data are contained within the article.

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
