# Peer review of "Assessment of the Safety Profile of Chimeric Marker Vaccine against Classical Swine Fever: Reversion to Virulence Study"

_viruses, 2024, doi:10.3390/v16071120_

Round 1

Reviewer 1 Report (Previous Reviewer 3)

Comments and Suggestions for Authors

Authors attempt to assess the safety profile of two chimeric marker virus ((vGPE− /PAPeV Erns or 154 vGPE− /PhoPeV Erns) or the parental vGPE− w), in particular regarding the reversion to virulence study, by passaging through piglets.

Although the two chimeric virus are demonstrated largely safe by far, reversion to virulence is still possible as demonstrated in reference 19.

Further study suggestions include:

1. extend a few more passages even after the negative detections (two passages, figure 1) to meet the life expectancy of  breeder pigs.

2. to consider the possibility of latent infection by chimeric virus.

3. In this study, the animal experiments were  conducted in laboratory containing situation, future test in the field  or more vivid situation is encouraged.

Author Response

Dear Reviewer 1,

Thank you for your time spent reviewing our manuscript. Please find an attached PDF of our response.

Best regards,

Reviewer 2 Report (Previous Reviewer 2)

Comments and Suggestions for Authors

I do not have any additional questions or comments additionally to my previous review and I recommend to accept manuscript in present form.

Author Response

Dear Reviewer 2,

Thank you for your time spent reviewing our manuscript. Please find an attached PDF of our response.

Best regards,

Reviewer 3 Report (Previous Reviewer 1)

Comments and Suggestions for Authors

.

Author Response

Dear Reviewer 3,

Thank you for your time spent reviewing our manuscript. Please find an attached PDF of our response.

Best regards,

This manuscript is a resubmission of an earlier submission. The following is a list of the peer review reports and author responses from that submission.

Round 1

Reviewer 1 Report

Comments and Suggestions for Authors

I reviewed the manuscript entitled :” Assessment of the safety profile of chimeric marker vaccine against classical swine fever”

In this study authors evaluate the safety profile of two vaccine candidates for CSF by conducting a reverse to virulence study based on WOAH recommendations.

Although the study provides relevant information regarding the safety profile of these vaccine candidates, including genetic information and ability of induce interferon, in my opinion authors failed to follow the WOAH recommendations to conduct the reversion to virulence study. The WOAH protocol indicates that once virus is not detected in 2 piglets, and additional passage in two pigs must be conducted. One additional passage was missed for the two vaccine candidates.  Authors must conduct and additional passage to comply the WOAH requirements.

Author Response

Dear Reviewer 1,

Please find attached our revisions based on your suggestions.

Thank you very much for your consideration.

Best regards,

Reviewer 2 Report

Comments and Suggestions for Authors

Based on my level on knowledge in this research area I do not have significant comments of questions to the manuscript and recommend to accept in present form.

Author Response

Dear Reviewer 1,

Please find attached our revisions based on your comments.

Thank you very much for your consideration.

Best regards,

Reviewer 3 Report

Comments and Suggestions for Authors

This is an extension of authors' previous study to assess the safety profile of two chimeric marker vaccine derived from the GPE-1 vaccine, which have been extensively used in the field in Japan and has proved successful in protecting against CSF.

The introduction can stop at line 93.  By here, readers should sufficiently know the purpose of the study.

section 2.7: Figure 1 should be cited here.  It would be difficult to understand, without referring to  figure 1, for the study design.

section 2.10 the ethical statement can be merged with section 2.6. Animal uses.

Table 1: The SNT used here is luciferase-based, which is presumedly more sensitive than the traditional SNT based on reading under the phase contrast microscope.  Please discuss if by day 7 poi, SNT <1, how can you demonstrate its protection efficacy, since IFN assay cannot be used routinely in the field?  Also, the negative virus recovery in two passages  is good, but to be used in breeding stocks, like sows and boars, you have to test it for at least 11 or 12 passages (to rule out latency), because the life expectancy for the breeding stocks is at least 6 times higher than fattening pigs.  

Figure 2 is good, but table 2 overlap significantly with it.  Consider merge table 2 into Figure 2.

Figure 3b: here the higher IFN-induction is "PhoPe" not "PAPe". Please confirm the labeling on the figure 3b.

Discussion

lines 330-356: there is already sufficient introduction on the studied issue in lines 52-66. So here a few words of reiteration is enough, you can go directly into line 357 after a few words.

Author Response

Dear Reviewer 3,

Please find attached our revisions based on your suggestions.

Thank you very much for your consideration.

Best regards,

Round 2

Reviewer 1 Report

Comments and Suggestions for Authors

.

Reviewer 3 Report

Comments and Suggestions for Authors

The R1 version has improved over the original version.